# Cytoskeletal Control and Wnt Signaling—APC’s Dual Contributions in Stem Cell Division and Colorectal Cancer

**DOI:** 10.3390/cancers12123811

**Published:** 2020-12-17

**Authors:** M. Angeles Juanes

**Affiliations:** 1School of Health and Life Science, Teesside University, Middlesbrough TS1 3BX, UK; m.juanes@tees.ac.uk; 2National Horizons Centre, Teesside University, 38 John Dixon Lane, Darlington DL1 1HG, UK

**Keywords:** APC, cell migration, gut renewal, colorectal cancer, stem cells, cell division, spindle orientation, Wnt signalling, actin, microtubules, cytoskeleton

## Abstract

**Simple Summary:**

Colorectal cancer is the third leading cause of cancer death globally. As well as the adverse health implications for the individual, there are also considerable economic and social impacts associated with workplace absence and healthcare expenses. It is critical to understand the events that govern gut homeostasis to improve cancer therapies. Intestinal cells proliferate and give rise to progenitor cells that then differentiate and actively crawl up to the villus tip where they are shed off. A balance between cell division and active migration is key to epithelium renewal. Alterations in the tumour suppressor Adenomatous polyposis coli (APC) have been found in more than 85% of colorectal cancer cases. Here we offer a perspective of APC’s dual roles—cytoskeletal hub and Wnt inhibitor—for their combined impact on gut epithelium maintenance and their dysfunction leading to cancer.

**Abstract:**

Intestinal epithelium architecture is sustained by stem cell division. In principle, stem cells can divide symmetrically to generate two identical copies of themselves or asymmetrically to sustain tissue renewal in a balanced manner. The choice between the two helps preserve stem cell and progeny pools and is crucial for tissue homeostasis. Control of spindle orientation is a prime contributor to the specification of symmetric versus asymmetric cell division. Competition for space within the niche may be another factor limiting the stem cell pool. An integrative view of the multiple links between intracellular and extracellular signals and molecular determinants at play remains a challenge. One outstanding question is the precise molecular roles of the tumour suppressor Adenomatous polyposis coli (APC) for sustaining gut homeostasis through its respective functions as a cytoskeletal hub and a down regulator in Wnt signalling. Here, we review our current understanding of APC inherent activities and partners in order to explore novel avenues by which APC may act as a gatekeeper in colorectal cancer and as a therapeutic target.

## 1. Introduction

Mitotic cell division is the fundamental process for eukaryotic cell duplication. Cell division may proceed mostly in two alternative ways: (i) Symmetrically to generate two cells with equal cellular contents and identity, or (ii) asymmetrically to generate two daughter cells with different size and/or developmental potential [1,2,3,4]. Of outstanding significance are asymmetric stem cell divisions in which one daughter cell retains its mother’s stemness while the other daughter is set on course to differentiate. This balance between self-renewal and differentiation, contributed in part by such a mode of asymmetric division, is critical for sustaining tissue homeostasis throughout the entire life of an organism. Perturbation of this delicate equilibrium can result in tumourigenic overgrowth and/or tissue degeneration [5,6,7].

It is the orientation of the mitotic spindle that determines the position of the cell division plane [8,9,10,11,12]. The mitotic spindle is a complex three-dimensional dynamic machine built by cytoskeletal components—microtubules (polymers of alpha/beta-tubulin subunits) and associated regulatory proteins [13,14,15]. Spindle assembly proceeds through multiple overlapping pathways (reviewed in [16,17]). Briefly, in most animal cells, microtubules are nucleated from the separating centrosomes and/or the vicinity of chromatin. Centrosomes define the spindle poles and promote bipolarity. In turn, bioriented attachment of sister chromatids ensures accurate chromosomal segregation into daughter cells. Many associated molecules help create this dynamic structure, including microtubule-associated proteins (MAPs), molecular motors, other microtubule-tracking proteins and cell cycle regulators [18]. In addition, the actin cytoskeleton appears to restrict microtubule growth around the centrosome during mitosis [19,20,21,22].

Spindle orientation responds to multiple layers of regulation that integrate cytoskeletal elements and cell cycle controls, including the role of cyclin-dependent kinases (CDKs), CDK inhibitors (CKIs), mitotic kinases, phosphatases and ubiquitin ligases (reviewed in [23,24]). In addition to those controls, extrinsic or intrinsic polarity cues determine the correct alignment of the mitotic spindle with respect to an axis of cell polarity. Many investigations carried out in asymmetric cell division models based particularly on yeast, flies and mammals show that the core machinery of spindle regulators is evolutionary conserved [1,10,25,26,27], despite some idiosyncrasies among those models. It is, therefore, an ongoing effort to consolidate lessons derived from these various lines of work into a set of fundamental principles that govern spindle orientation in cells dividing asymmetrically and offer insight into outstanding questions. 

One of these open questions relates to the mechanisms implicating the tumour suppressor Adenomatous polyposis coli (APC) in tumourigenesis, in view of its dual involvement in signalling (Wnt pathway [28,29,30,31,32,33,34,35,36,37,38,39]) and cytoskeletal function (with ties to both actin and microtubules [38,39,40,41,42,43,44,45,46,47,48,49,50,51,52,53,54,55,56,57]). Of note, in the literature, there is not a clear consensus in the nomenclature to refer to the APC as a gene and protein. For simplicity, in this review, hereinafter, *Apc* denotes the gene and APC the protein. However, nomenclature for individual mutants follows the common practice in the literature. *Apc* has been found mutated in ~85% of colorectal cancer cases [28,58,59], and heterozygous mutations result in spindle misorientation and altered cell shape [43,44,45,46,60,61,62]. However, a definitive link between spindle orientation in stem cell divisions and gut homeostasis remains unclear, and thus, whether cytoskeletal APC dysfunction might drive the path to cancer. Here we offer an integral view of the possible contributions of APC in intestinal stem cell self-renewal, cell migration and gut homeostasis.

## 2. The Tumour Suppressor *Apc*, a Master Regulator of Gut Homeostasis 

### 2.1. Cellular Organisation in the Gut—From Intestinal Stem Cells to Differentiated Cells 

The intestine represents the most vigorously renewing organ in mammals. The intestinal epithelium is organised into highly proliferative invaginations called crypts of Lieberkühn and differentiated finger-like villus structures (Figure 1) [63,64]. Self-renewing stem cells are located at the bottom of the crypts and are critical to sustaining normal tissue architecture [65]. Stem cells give rise to Paneth cells, which accumulate at the bottom of the crypts, close to the stem cells and together occupy the microenvironment known as the stem cell niche. Due to their importance in homeostasis, defining the exact localisation and properties of the intestinal stem cells has been the subject of intensive effort. At least two pools coexist in the crypts (Figure 1; reviewed in [66,67,68,69,70,71]). The best characterised are actively dividing stem cells at the bottom of the crypts, known as columnar base cells (CBCs), which are interspersed with Paneth cells [67,72,73]. Another stem cell pool resides immediately above the uppermost Paneth cells, on average at the +4 position of the intestinal crypt. These ‘label-retaining cells’ (LRCs) maintain the incorporation of tritiated-Thymidine or bromodeoxyuridine (BrdU)—an analogue of the nucleoside thymidine—in pulse-label DNA experiments. By contrast, transit-amplifying cells divide fast and, as a consequence, dilute out the label. LRCs remain quiescent and act as a reserve population that can give rise to all intestinal cell lineages after tissue damage [74,75,76,77]. A variety of molecular markers have been assigned to the different cell types, including the most common marker for CBC stem cells—the leucine-rich-repeat-containing G-protein-coupled receptor 5-expressing, Lrg5+ (Figure 1). 

Paneth cells secrete bacteriostatic molecules to protect the stem cells [78,79,80,81]. In addition, stem cells produce a group of partially differentiated cells (known as transit-amplifying cells), which mostly stay in the same position, but after five to seven additional divisions move upwards and fully differentiate into various lineages that populate the villus (Figure 1) [82,83,84,85,86]. Fully differentiated cells are: (i) Enteroendocrine cells (scattered throughout the epithelial layer and release hormones); (ii) enterocytes (absorptive epithelial cells in the small intestine facing the gut lumen and with extensive microvilli at their apical surface); (iii) goblet cells (secrete mucus required for the effective movement, diffusion of gut contents, and protection against shear stress and chemical damage). These fully differentiated cells actively migrate along the crypt-villus axis towards the tip of the crypt [87], where they will undergo apoptosis and eventually shed off over a cycle lasting 3–5 days in humans [88]. This fast cell shedding is counterbalanced by the proliferation of the stem cells and the progenitor cells.

### 2.2. Overview of Signalling Pathways Controlling Gut Epithelium Homeostasis and Relevance to Therapeutics

The existence of multiple gradients of signalling based on Wnt, Bone morphogenetic protein (BMP), Notch and epidermal growth factor (EGF), among others, promote correct epithelium architecture (Figure 1) [85,89,90]. Paneth cells secrete Wnt, EGF and BMP antagonists, and those gradients present higher concentration at the bottom of the crypt; while BMP signalling is active in the villus compartment. Negative cross-talk between Wnt and BMP signalling controls stem cell expansion. The intestinal stem cell compartment and enterocytic cell fate are additionally controlled by the interplay between Wnt and Notch [91]. Notch stimulates both stem cell maintenance and cell fate decision between absorptive and secretory cells, and can induce Paneth cells to de-differentiate upon tissue damage [92]. Hyperactive Notch blocks the commitment of cells to adopt a secretory lineage fate [93]. Conversely, Notch inhibition causes loss of CBCs, pointing to its requirement for stem cell proliferation and survival [94].

Wnt gradient is arguably the most critical gradient [95,96,97,98,99,100,101,102]. This gradient has also been known as the counter-current-like mechanism in the crypt because of its inverse correlation to APC concentration (which is low at the bottom of the crypt) [103]. Hyperactive Wnt signalling promotes tumourigenesis and may interfere with anticancer drugs [102]. Conversely, Wnt inhibition either by Dkk1 or genetic disruption of the Wnt pathway effectors Tcf4 or beta-catenin, promotes stem cell quiescence [73,100,104,105,106]. Regarding therapeutic prospects, Wnt remains a prime target (reviewed in [107,108,109,110,111,112]). For example, Tankyrase inhibitors belonging to the poly(APD-ribosyl)ases have been used to inhibit Wnt/beta-catenin activity [113,114,115,116]. However, the success of the therapy is highly dependent on the specific *Apc* mutation [114]. For instance, mutants affected in the central region of *Apc* respond to Wnt inhibition by Tankyrase, but not N-terminal mutants. Thus, genetic events can lead to profound changes in response to colorectal cancer therapies. Other chemotherapeutic agents widely used are taxanes (e.g., paclitaxel or docetaxel), which stabilise microtubules, disrupt mitosis and cell cycle progression [117]. However, taxane resistance has been a drawback [118]. Recent combinatorial approaches between taxanes and small molecules that inhibit Wnt seem to have the potential to combat the resistance by synergising mitotic blockade [108,118]. One of this Wnt inhibitors is ipafricept (OMP-54F28). This drug binds to Wnt via a cysteine-rich domain abolishing the interactions between Wnt and Frizzled receptors. The combination of the OMP-54F28 with paclitaxel and carboplatin looked promising in ovarian cancer patients, however, efficacious doses were associated with bone toxicity, halting future investigations [119]. The combination of OMP-54F28 with nab-paclitaxel and gemcitabine led to tumour regression in patients with pancreatic cancer [120]. However, these conclusions need to be taken with caution since a low number of patients have been successfully monitored [120]. Another Wnt inhibitor is Vantictumab (anti-FZD). Anti-FDZ blocks binding of Wnt to Frizzled receptors [108]. Sequential doses of anti-FDZ prior to intermittent taxane treatment led to an enhanced antitumour response in various cancers, but toxicity remains an issue [108]. In conclusion, despite drug combinations of taxanes with Wnt inhibitors representing a promising strategy for cancer therapies, further development of these drugs (e.g., safer dose and/or different formulation to avoid toxicity) remains a challenge [111,112,119,121].

### 2.3. Modes of Intestinal Stem Cell Division and Homeostasis

The mode of stem cell division in the gut is a controversial topic—both symmetric and asymmetric stem cell division are proposed to contribute to homeostasis. The division pattern of stem cells is regulated by intrinsic cell polarity cues, cell shape and/or environmental signals [122,123,124]. Biased spindle orientation in asymmetric cell divisions will either couple intrinsically the orientation of the division plane with the asymmetric segregation of cell fate determinants or change the respective surroundings encountered by the two daughter cells [62,125,126,127]. A study showed that in both mouse and human gut wildtype stem cells, the mitotic spindle orients predominantly perpendicular to the apical surface, whereas the spindle oriented in an unbiased manner, in transit-amplifying cells [62]. By measuring the spindle orientation angle along with the retention of DNA marks to monitor gut stem cell pools, a correlation was found between perpendicularly aligned mitotic spindles and asymmetric segregation of labelled DNA. However, precancerous cells exhibited random spindle orientation (that might be partly attributed to their rounder shape), alongside random DNA marking. This supports the idea that stem cell division would be inherently asymmetric, a pattern lost in cancer cells. However, this model has been challenged by lineage tracing studies proposing that cell division follows a stochastic pattern described as neutral drift dynamics, in which Lrg5+ divisions are predominantly symmetric and yield two cells with the same fate that potentially compete for space within the niche [128]. Yet, more recently, a counter-proposal has favoured instead an asymmetric cell division-dominant neutral drift model also validated in vivo [129]. These contrasting findings could be related to experimental details, e.g., nutrient availability, harsh environment or aging [124,130]. Indeed, the frequency of self-renewing asymmetric division changes as a function of age to respond to developmental or local effects as well as stress associated with micro-injuries both in mice and flies [7]. Taken together, the extent of symmetric versus asymmetric divisions supporting gut homeostasis are starting to emerge. 

### 2.4. Apc as the Gatekeeper Gene in Colorectal Cancer 

The rapid clearance of gut epithelial cells ensures that potentially oncogenic mutations, which can occur with some frequency in the hostile stressful environment of the gut, do not prevail in promoting tumourigenesis. However, oncogenic mutations arising in cells within the stem cell niche may drive the path towards cancer. In colorectal tumours the first oncogenic mutation provides a selective advantage to the epithelial cell that proliferates and generates a clump of cells known as a microadenoma or adenomatous polyp. 

According to data from ‘The Cancer Genome Atlas Project’ (TCGA) available at the National Cancer Institute Genomic Data Commons (https://portal.gdc.cancer.gov, [131]) and the cBioportal for cancer (https://www.cbioportal.org, [132,133]), in >80% of human colorectal cancers, initiator mutations occur in the *Apc* gene (Figure 2). These are mainly autosomal-dominant mutations most of which result in protein truncations eliminating the C-terminus (Figure 2) [28,134,135,136]. Additionally, *Apc* mutations have been found in 5–20% of other types of cancer, e.g., stomach, uterine, ovarian and triple-negative breast cancer [137,138,139,140,141]. Why are *Apc* mutations so frequent in colorectal cancer cases and not in other cancer types? That is an intriguing question. One speculation is that tissue-specific features may render differential sensitivity to loss of *Apc*; perhaps the fast turnover of the gut epithelium explains the prevalent association between *Apc* inactivation and colorectal cancer. For space reasons, this review has focused on *Apc* involvement in gut homeostasis and its relevance to colorectal cancer [142], both highly conserved across species [142,143,144]. Indeed, loss of *Apc* in the murine, zebrafish and *Drosophila* gut recapitulates many aspects of the human disease. [143,144] Importantly, it has been proposed that *Apc* may be a marker for adult gut stem cells in *Drosophila* [143]. 

Germinal mutations in *Apc* are responsible for a hereditary syndrome known as Familial Adenomatous Polyposis (FAP) or familial polyposis coli (FPC) [145,146,147,148,149,150,151]. FAP patients develop hundreds to thousands of colon polyps, which progress towards malignancy through ordered histopathological stages that include high-grade dysplasia and invasive adenocarcinoma. Mutations in *Apc* are also responsible for the majority of sporadic colorectal cancers [46]. In both inherited and sporadic cancers, alterations in *Apc* lead to highly proliferative gut cells that fail to differentiate or migrate up the crypt-villus axis, lying dormant while harmless [58]. Less than 10% eventually, and over the years, will pick up subsequent inactivating mutations in other tumour suppressor genes such as TP53 phosphatase and tensin homolog (PTEN), mothers against decapentaplegic homolog 4 (SMAD4) as well as oncogenic mutations, e.g., Kirsten rat sarcoma viral oncogene homolog (KRAS), and phosphatidylinositol-4,5- bisphosphate 3-kinase, catalytic subunit alpha (PIK3CA). These mutations, followed by clonal expansion and epigenetic changes, promote progression into more aggressive tumours by invading the basement membrane, and eventually spreading to distant organs (Figure 2) [36,152,153,154,155]. 

Reintroduction of full-length APC in a SW40 colorectal cancer cell line that expresses truncated APC restores normal morphology, cell-cell adhesion and migration in wound healing assays [156]. More importantly, in a transgenic mouse model with colorectal cancer, restoration of APC protein levels promoted differentiation and reverted normal crypt homeostasis, even in the presence of hyperproliferative polyps due to alterations in KRAS (activating alteration) or TP53 (inactivating alteration) [157]. In addition, APC restoration promoted disease regression in the gut [157]. Moreover, restoration of APC activity has also shown a potential benefit in patients with inherited colorectal cancer carrying nonsense mutations that lead to premature stop codons [158]. As a proof-of-concept, treatment with the antibiotic erythromycin resulted in the restoration of APC activity, a decrease in adenoma size and burden in a pool of colorectal cancer patients. Similarly, induced read-through nonsense mutations using aminoglycosides and macrolides (e.g., tylosin) in mice carrying *Apc* nonsense mutations led to improved APC protein levels and clinical symptoms (e.g., reduced polyp size and extended life span) [159]. Furthermore, a truncated *Apc* can be selectively targeted by TASIN-1 [160]. TASIN-1 inhibits cancer cells grown in human xenografts and in a genetically engineered mouse model of colorectal cancer without affecting Wnt activity and presented low toxicity. Considering the high prevalence of *Apc* mutations in patients with colorectal cancer and that some mutations affect the APC basic domain, targeting specific sites on APC and/or activities should help to understand the molecular details of largely unexplored APC cytoskeletal interactions bringing crucial insights into cancer therapies. Given the central importance of APC in colorectal cancer and its potential as a therapeutic target, it is of great interest to achieve a comprehensive understanding of the multiple mutations in *Apc* and their functional impact at the molecular level.

## 3. APC Functional Domain Analysis and Animal Models for Investigation of Colorectal Cancer

APC is a large (2843 amino acid) multi-functional protein acting as a homodimer. APC interacts with multiple partners in vivo (Figure 3) [38,161]. Domain analysis has outlined (a) an N-terminal region of APC (residues 1–958) including three domains that mediate self-association, an Armadillo repeat region that interacts with IQGAP, ASEF, other cytoskeletal regulatory proteins like Kap3, and intermediate filaments; (b) a central region (residues 959–2129) containing three 15-amino acids repeats and seven 20-amino acid repeats that bind beta-catenin, three Serine-Alanine-Methionine-Proline (SAMP) motifs that bind Axin and two nuclear localisation signal (NLS); (c) a C-terminal region (residues 2130–2843), harbouring the so-called basic domain (2167–2674), which binds various cytoskeletal components including microtubules, the kinesin-1 mitochondrial adapters Miro and Milton, formins (e.g., Dia1 and Daam1), actin monomers, and the microtubule plus end-binding protein (EB1), which also binds the tail region of APC (2675–2843) [38,41,51,52,53,55,57,162,163,164]. 

The central region (amino acids 959–2129) has been extensively studied because it is critical to down-regulate the Wnt signalling pathway [29,30,58,142,165,166]. APC forms together with Axin the core of the so-called beta-catenin destruction complex [167], which phosphorylates beta-catenin, targeting it for ubiquitination and consequent proteasomal degradation (Figure 4). Wnt stimulation induces inactivation and dissociation of the complex, resulting in stabilization of beta-catenin, which can enter the nucleus and act as a transcriptional coactivator, inducing proliferation and differentiation. 

The C-terminal region (amino acids 2130–2843) that interacts with a variety of cytoskeletal proteins is critical for cell polarity, directed cell migration and focal adhesion turnover as well as spindle and chromosome dynamics [37,43,45,46,50,53,54,56,58,60,61,142,168,169,170,171]. More than 60% of the mutations found in *Apc* are concentrated in the mutation cluster region (MCR), which is located within the beta-catenin binding domain (amino acids 1284–1580) (Figure 2 and Figure 3) [147,172]. *Apc* mutations in both inherited and sporadic colorectal cancer include point mutations, small deletions or insertions, often resulting in a premature stop codon. The corresponding loss of roughly half to three-quarters of the C-terminal region eliminates both Wnt signalling function and cytoskeletal interactions (Figure 3).

The protein sequence and domain structure in APC are evolutionary conserved between murine and humans with 89% identity and 91.9% similarity shared at the amino acid level [144]. This high conservation made murine widely used models investigate the role of *Apc* in colorectal cancer—at least 43 different mouse models and 3 rat models have been established [173,174,175,176] (Figure 3). The first genetically engineered mouse model, known as *Apc*^Min^ (for multiple intestinal neoplasia), contains a nonsense point mutation in the codon 850 that behaves as autosomal dominant loss of *Apc* function [177,178] and promotes intestinal adenomas (polyps). Homozygous *Apc*^Min^ mice die early in embryonic development [177,179]. Later mouse models consisting of deletions of specific *Apc* domains, e.g., the *Apc*∆SAMP (amino acids 1322 to 2005) or the *Apc*mNLS (which contains two internal deletions that disrupt both NLSs) have been used to confirm the impact of *Apc* on beta-catenin levels via Wnt signalling. Strikingly, a homozygous mice model containing a mutation at codon 1638 (*Apc*1638T) is viable and fertile, although it has developmental defects such as growth retardation and reduced postnatal viability [180]. Of note, this truncated protein retains 3 of the 7 20-amino acid repeats and 1 SAMP motif but lacks the rest of the C-terminus. Phenotypic studies using this model revealed that loss of sequences that lie C-terminal to the beta-catenin regulatory domain in *Apc* may contribute to chromosome instability in colorectal cancer, and, therefore, tumourigenesis, independent of Wnt signalling [46]. The reason for the relatively mild phenotype of *Apc*1638T remains unclear. However, a plausible interpretation is that there might be functional redundancy between APC and its homologue APC2 [180]. The phenotype of an *Apc* rat model supports this idea. *Apc* KAD rat or Kyoto *Apc* Delta contains a germline nonsense mutation (S2523X), which encodes a truncated protein lacking the C-terminal 321 amino acids. KAD rats did not develop spontaneous polyps, instead, they presented enhanced colitis-associated colorectal cancer. Altogether, these models resulting in deletions or truncated *Apc* have rendered the Wnt pathway hyperactive by disrupting beta-catenin control [31], which presumably contributes to tumourigenesis [28]. However, the additional impact of the loss of APC’s cytoskeletal function cannot be excluded, as inferred from the behaviour of certain C-terminal mutants that retain Wnt function. New animal models to design selective *Apc* deletions, e.g., use of the LoxP-Cre system or CRISPR genome-editing would help dissect the precise contributions of Wnt and cytoskeletal activities to gut homeostasis. Further, such mutants will shed light into the long-standing question of whether both of these APC activities are mechanistically connected [173,174,175]. 

## 4. Control of Cell Migration beyond Wnt Signalling—APC as a Cytoskeletal Hub

### 4.1. Cell Migration in the Gut

It has been long assumed that cells within the crypts and along the villi migrate passively, driven by the upward pressure resulting from cell division in the crypts—as cells divide, they push their neighbours along the crypt-villus axis [72,181]. However, abrogation of mitotic activity (e.g., after cell irradiation or pharmacological treatment) did not prevent such migration [182]. A recent study [86,87] based on experiments in mice and explants combined with mathematical modelling, demonstrated that intestinal cells actively crawled up from the crypt towards the villus tip, while pressure arising from mitotic division exerted a limited effect within the lower region of the villus [87]. This study showed that actin-rich protrusions along the villus-axis are critical to generate the force to move up to the villi surface. Those actin protrusions appear to be mainly generated by the actin-related protein 2/3 (Arp2/3) complex; which nucleates branched actin networks upon stimulation via Wiskott–Aldrich syndrome protein (WASP)/Scar and WAVE (WASP family veroprolin-homologue) [183,184,185]. Mice treated with CK666, a specific inhibitor of Arp2/3, or mice knocked out for one subunit of the Arp2/3 complex exhibited disrupted cell protrusions and migration, without any effects on cell adhesion or mitosis. In brief, this study proposes that Arp2/3-driven actin migration along the villus might contribute to gut renewal. Interestingly, studies based on platinum replica electron microscopy showed that branched actin networks are assembled by APC on microtubule tips to direct protrusions in neurons and fibroblasts-like COS7 cells [186]. Taken together, it is tempting to propose that APC may further participate in cell migration via actin-dependent protrusions in the gut.

### 4.2. APC, Cytoskeletal Dynamics and Cell Migration 

APC’s role in cell migration has been classically tied to mechanisms involving microtubule dynamics (directly or indirectly via microtubule regulatory proteins like its most famous molecular partner EB1) or via actin interactors [37,41,47,50,58,142,187,188]. More recently, it has been demonstrated that APC binds to actin and participates directly in actin cytoskeleton functions, including cell migration. In vitro pull-downs and single molecule, TIRF microscopy led to the outstanding observation that the same domain of APC that binds microtubules also binds actin and that APC potently nucleates actin filaments in vitro, alone or in collaboration with formins [51,52,55]. In addition, exploiting allele-specific APC mutants, this activity proved essential to drive microtubule-capture coupled to focal adhesion turnover, critical for directed cell migration [53,56]. The potential for combinatorial control is further demonstrated by the fact that APC-mediated actin nucleation is directly inhibited by EB1, independent of EB1′s interactions with microtubules [54]. 

In summary, (i) *Apc* is considered the master regulator of gut homeostasis, (ii) Arp2/3-driven actin migration contribute to gut renewal, (iii) APC displays direct interactions with microtubules, actin and microtubules and/or actin-regulatory proteins; (iv) APC-driven actin nucleation promotes directed cell migration but also appears to direct protrusions. Taken together, these observations raise the possibility that APC alone or in coordination with Arp2/3, and perhaps with formins and microtubules, finely orchestrates directed cell motility along the crypt-villus axis by inducing branching and/or filamentous actin-rich protrusions. The relevance of APC-mediated actin activity in the renewal of the epithelium and in disease is an exciting field awaiting to be explored. 

## 5. APC Roles in Spindle Morphogenesis and Dynamics

APC cytoskeletal roles may extend well beyond cell migration. Pioneer work in *Drosophila* male germline stem cells showed that APC controls spindle orientation to sustain stem cell division and differential cell fate [189]. The involvement of APC in spindle orientation has proved conserved among a variety of models. Besides spindle orientation, APC has been implicated in other aspects of spindle and chromosomal dynamics, all bearing relevance to the impact of *Apc* inactivation on tumourigenesis.

### 5.1. APC at the Kinetochore-Microtubule Interface and Chromosomal Instability

Genetic and chromosomal instability are regarded as enabling characteristics in cancer [190]. EB1 and APC are localised independently at microtubule plus ends and kinetochores. Plus-end tracking may control microtubule dynamicity and, in addition, spindle orientation (see below). 

Depletion of *EB1* or *Apc* perturbs both chromosome attachments and congression and increases the number of lagging chromosomes [43,61,191]. This indicates that an APC-EB1 complex might provide a physical link between microtubule growing ends and kinetochores [43,61,191]. The precise binding sites or the underlying mechanism(s) controlling APC-EB1 function at the kinetochores remain unclear but might involve posttranslational modifications. Indeed, APC undergoes phosphorylation during mitosis by human p34(cdc2)-cyclin B1 in HCT116 colorectal cancer cells [192]. This modification downregulates the association between APC and EB1, thus acting as a switch controlling the APC-EB1 complex [164]. APC’s localisation and function at kinetochores are important for accurate chromosome segregation [43,46,170]. In the *Apc*^Min^ and *Apc*1638T mouse models [177,180], inefficient microtubule-kinetochore attachments were observed during mitosis, which led to chromosomal instability [46]. Some studies have explored the relationship between APC at kinetochores and the spindle assembly checkpoint (SAC). This surveillance mechanism monitors sister kinetochore attachment and halts the cell cycle in metaphase upon errors [193]. Defects on the SAC could promote tumourigenesis on their own [190]. However, some reports showed that *Apc* knockdown or knockout activates the SAC, while others showed that they impair the SAC (for specific details, see [43,44,45,61,191,194,195,196,197]). Despite the precise relationship between APC and the SAC remains unclear, evidence points to a clear requirement of APC to avert chromosomal instability, thus suggesting an alternative contribution of *Apc* inactivation toward tumourigenesis.

### 5.2. Cortical APC in Spindle Orientation and Cell Migration 

APC localised at cortical microtubules plays a variety of roles. APC stabilises interphase microtubules in contact with the cell cortex during directed cell migration [40,43,45,50,198,199,200]. APC requirement for correct spindle positioning may be inferred from the impact of *Apc* mutants on astral microtubule-cortex interactions. Indeed, *Apc*^Min^ and *Apc*1450 disrupt astral microtubule function leading to spindle positioning defects—spindles placed off-centre and exhibiting excessive rotation. Additionally, these *Apc* mutants cause cytokinetic failure [44]. *Apc* knockdown in interphase U2OS cells delays microtubule regrowth following nocodazole wash-out [201,202]. To date, mitotic roles for APC at centrosomes are likely, but those have not been demonstrated.

Interestingly, the centrosome appears to be an actin-organising centre [22]. Actin restricts microtubule growth at the centrosome to organise the mitotic spindle [19,20,21,22]. Indeed, Arp2/3-complex at the centrosome induces actin nucleation near the centrosome that appears to decrease microtubule density during prometaphase in HeLa cells [203]. A pool of F-actin also dependent on the Arp2/3 complex around the mitotic spindle has been observed during anaphase in HeLa cells [21]. In Xenopus embryonic epithelia, at least two pools of actin seem to associate with the mitotic spindle: (i) Stable actin cables linked to the mitotic spindle, (ii) finger-like long actin cables that transiently extend from the cell cortex and target the mitotic spindle, being these interactions sensitive to the SMIFH2 formin inhibitor [204]. Furthermore, in budding yeast, a unifying principle is that the interaction between Bud6 (a protein with dual connections with actin and microtubules) and Bim1 (the budding yeast EB1) is required for cortical capture of microtubules [205]. In addition, actin contributes to orient the mitotic spindle either serving as rides for Kar9 (the proposed functional counterpart of APC) on motor proteins to reach the cortex and/or through Bud6, which binds G-actin and formins to polymerise actin, and it is key to capture microtubules at the cell cortex via EB1 interactions [205,206,207,208]. The triad of mammalian proteins APC, EB1 and formin could play a similar role in spindle orientation and astral microtubule capture. 

APC nucleates F-actin by itself or synergistically with formins [51,52,53,55,56]. Importantly, APC-mediated actin nucleating activity (with or without formins) is inhibited by EB1 [54]. On these premises, an attractive possibility is that APC subject to modulation by EB1 could generate actin filament networks to help stabilise and orient the spindle. Furthermore, actin filaments perhaps formed in collaboration with formins, may be exchanged between the cell cortex and the mitotic spindle. Thus, cortical APC may be crucially at a cross-road controlling actin function to orchestrate force generation required for migration of differentiated cells along the crypt-villus axis, in addition to any contributions to spindle orientation in proliferating cells at the base of the crypt. With regards to cell migration, these actin structures may act as the yaktrax ice grips under your shoes to reduce the risk of slips when climbing an icy hill. Although various aspects of this hypothesis await testing, that APC plays a critical role in spindle orientation in gut stem cell division has been demonstrated. Indeed, loss of APC function impairs biased spindle orientation at the stem cell niche (Figure 1) [59]. 

## 6. Conclusions and Future Perspectives

APC is a multidomain protein with numerous functions in cell division, cell migration and gut epithelium renewal. The central part of APC mediates downregulation of Wnt signalling, and the N- and C-terminal regions have been linked to cytoskeletal regulation with an impact on cell migration and spindle orientation. It is the C-terminal part that interacts directly with actin, microtubules and actin/microtubule regulatory proteins. The fact that most of the key cytoskeletal interactions are concentrated on the C-terminal basic domain of APC makes it difficult to disentangle specific activities within this cytoskeletal hub. Indeed, interpretation of phenotypes arising from assays using null, deletion or truncation mutants of APC remains inconclusive. The only way forward is to probe APC activity by generating separation-of-function mutants. One specifically disabled in actin-polymerising activity—APC-m4 [53], provided direct evidence for the essential role of APC via actin in directed cell migration and focal adhesion turnover. This mutant’s impact on gut homeostasis is currently unknown. However, this approach should be expanded to produce a complete mutant toolkit to decipher the significance of the interactome centred on APC. Finally, with methodologies that offer unprecedented access to explore gut epithelium organisation in 3D it should be possible to achieve an integral understanding of the molecular basis for APC central role in gut homeostasis (Wnt-dependent and independent mechanisms). This knowledge will represent a positive step to find novel therapeutic regimes that selectively and efficiency target Wnt signalling and/or *Apc* to re-engage endogenous mechanisms and functions restoring normal homeostasis and that hopefully evoke side effects rarely. Further, identification of tumour-specific driving mutations may have an impact on personalised cancer therapies. 

## Figures and Tables

**Figure 1 cancers-12-03811-f001:**
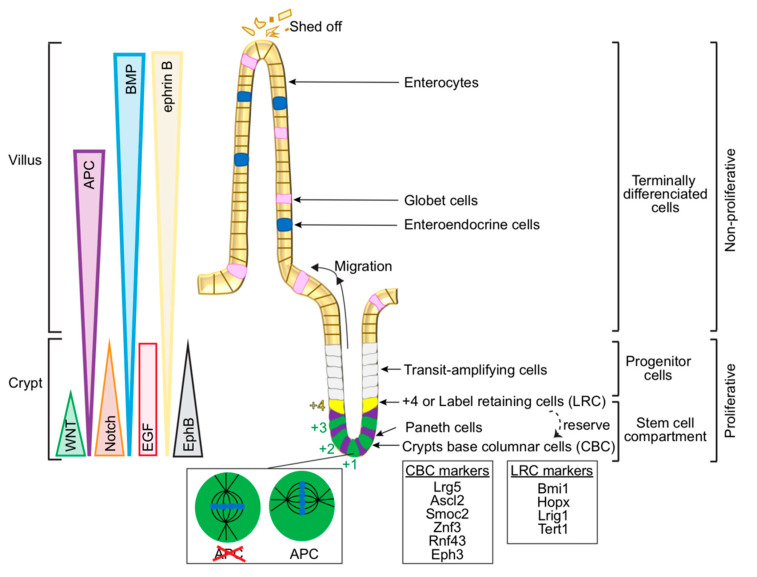
Small intestinal epithelium organisation and signalling pathways. Schematic of the gut epithelium depicting the signalling gradients that control cell homeostasis, as well as cell types, including stem cell populations—the crypt base columnar cells (CBC) at the bottom of the crypt and the label-retaining cells (LRC) at the +4 position, and respective markers. Zoom into a stem cell at the bottom of the crypt (+1), showing *Apc*-dependent orientation of the mitotic spindle. According to [62], the mitotic spindle orients preferentially perpendicular to the apical surface in the presence of *Apc* but favours a parallel orientation in the absence of *Apc*. Brackets define the cellular compartments containing proliferative cells (stem cells and transit-amplifying cells), and non-proliferative cells (terminally differentiated cells—enteroendocrine cells, globet cells and enterocytes. Differentiated cells actively migrate up to the top of the tip villus where they shed off.

**Figure 2 cancers-12-03811-f002:**
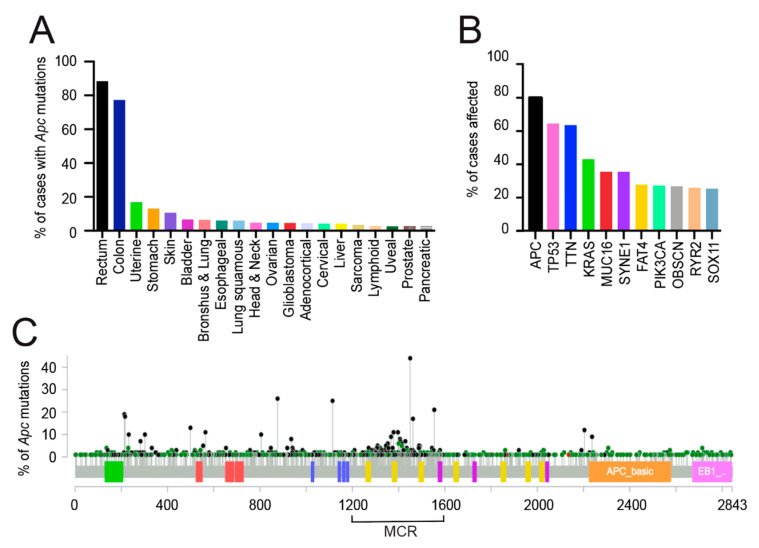
Frequency of *Apc* mutations based on ‘The Cancer Genome Atlas Project’ (TCGA) from the National Cancer Institute Genomic Data Commons and the cBioportal databases [131,132,133]. (**A**) Percentage of cases with *Apc* mutations found across the different cancer projects. The primary sites with most *Apc* mutations are the rectum (black bar) and the colon (blue bar), as retrieved from the National Cancer Institute Genomic Data Commons database. (**B**) Mutation frequency of the indicated genes derived from the TCGA—Rectum and Colon projects. (**C**) Lollipop plot showing *Apc* mutations annotated in TCGA from the cBioPortal database. Green circles represent missense mutations (471 in total) and black circles represent truncating mutations (751 in total). The mutated cluster region (MCR), where most of the *Apc* mutations occur, encodes the region spanning amino acid position 1284 to the 1580. The most frequent mutations introduce a stop codon at amino acid position 1450 (AA change to R1450* (stop)). Colour code boxes indicate various domains in APC (see text).

**Figure 3 cancers-12-03811-f003:**
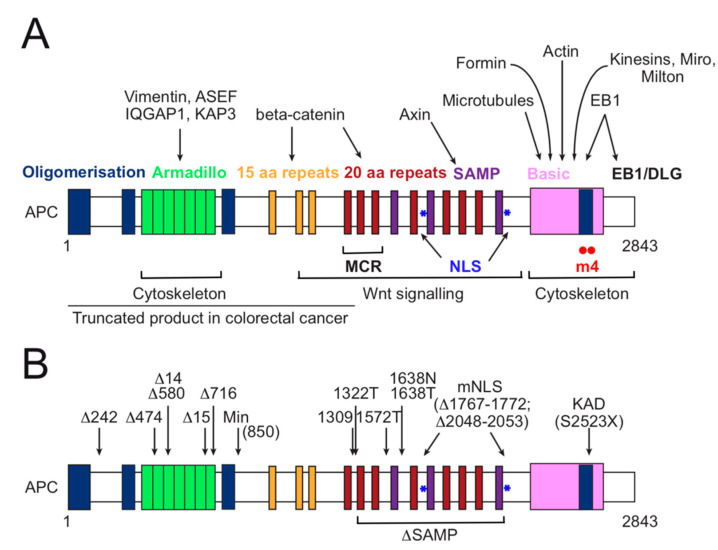
Human APC protein structure and sites (mutated or truncated) in murine models. (**A**) Schematic showing the position of APC domains (coloured boxes and asterisks) (from left to right): Oligomerisation, Armadillo repeats, 15 and 20 amino acid repeats, Serine-Alanine-Methionine-Proline (SAMP) motifs, nuclear localisation signal motifs (NLS), basic domain (which contains one oligomerization domain shown in blue), and the end-binding protein (EB1)/discs large (DLG) domain. Arrows link APC partners to the corresponding interaction sites. The mutated cluster region (MCR) corresponds to amino acids 1284 to 1580. Blue asterisks indicate the location of the two NLS, which comprise to amino acids 1767–1772 and 2048–2053. Red dots indicate the location of the APC-m4 mutation (L2539A I2541A) in the oligomerization motif within the basic domain. In most colon cancers, mutations lead to a truncated product lacking both signalling and cytoskeletal functions. (**B**) Schematic indicating the position of the germline *Apc* mutations in various murine models. The names of the mutations refer to the common name used to describe a particular murine cancer model. Δ indicates deletion; AA, amino acid (number in the sequence and change); Min (Multiple intestinal neoplasia), mNLS (mutant in nuclear localisation signals), KAD (Kyoto *Apc* Delta).

**Figure 4 cancers-12-03811-f004:**
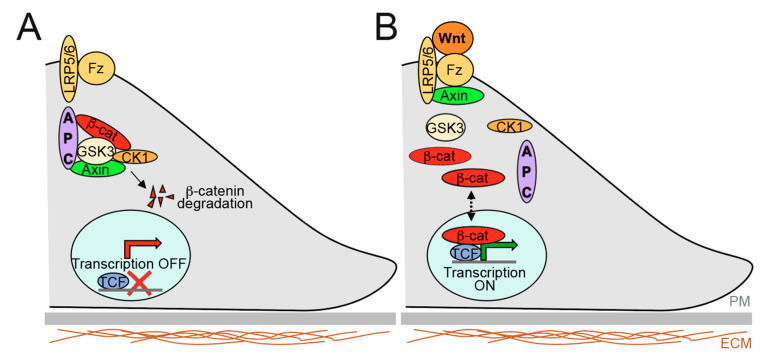
General view of Wnt signalling pathway. (**A**) In absence of Wnt signalling, APC forms a ’destruction complex’ with Axin, Casein Kinase (CK1), Glycogen Synthase Kinase 3 (GSK3) that leads to hyperphosphorylation of beta-catenin, and consequently degradation. (**B**) Wnt stimulation leads to the inactivation and/or dissociation of the destruction complex. In this case, Wnt ligand binds to a Frizzled (Fz)/LRP5/6 cell-surface receptor complex. This Wnt complex leads to stabilisation of hypophosphorylated beta-catenin, which enters the nucleus and interacts with transcription factors (TCF/LEF proteins) to activate gene transcription. PM: Plasma membrane. ECM: Extracellular matrix.

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
