# Peer review of "Cytoskeletal Control and Wnt Signaling—APC’s Dual Contributions in Stem Cell Division and Colorectal Cancer"

_cancers, 2020, doi:10.3390/cancers12123811_

Round 1
Reviewer 1 Report
In the present work the author summarizes the role of APC as multidomain protein and describes their numerous functions in cell division, cell migration and gut epithelium renewal. Author focuses in APC dual roles as cytoskeletal hub and Wnt inhibitor, and the impact on gut epitheliym maintenance and dysfunction leading to cancer. Only a few minor issues should be addressed:
MINNOR POINTS:
- I suggest to include a section (or at least comment) explaining the potential role of APC in therapeutic strategies, due to that some published data indicate that the re-expression (or restauration) of APC, both in vitro and in vivo models, functions as a therapeutic strategy in some cancers including colon cancer. In my opinion this new section could improve the quality of this complete review, and have a strong connection with the information described in the actual version of the manuscript regarding the role of APC as suppressor of the canonical WNT signaling pathway.
- In Figure 2, a list of abreviation used in Fig 2A should be included.
Reviewer 2 Report
Manuscript ID: cancers-1024927
Cytoskeletal control and Wnt signalling — APC's dual contributions in stem cell division and cancer
The manuscript by Angeles Juanes is a timely, enjoyable and interesting review on an important subject. The author well illustrates the relevant literature and the manuscript is well written.
Nevertheless, before acceptance I want to itemize only minor points to be checked by the author.
- The nomenclature for “APC” is inconsistent in this manuscript (Apc, Apc, APC, APC). Please correct. Please distinguish only APC protein vs. Apc gene (human and murine).
- Since Figure 2c is already colored, I suggest to use also colored figures for 2a and 2b.
- Line 185: The definition of the manuscript “APC” terminology should be visible much earlier in the manuscript (and not on page 5).
- Line 269: “(with contains mutations 269 that inactivate both NLSs)” must be corrected.
- Please check whole manuscript for consistency: In vitro/in vitro in vivo/in vivo
Reviewer 3 Report
This is an academic review article related to the role of APC in Stem cell regulation by virtue to Wnt signaling and cytoskeletal control.
This article is mostly academic application and much less for cancer therapy or tumor progression aspect.
Critical comments:
- Too lengthy
- Make it short, crispy and cancer oriented, since it is basically a cancer related journal (not a basic science journal)
- Stem cell part is way bigger and CRC part is way less (it is the most common altered gene in CRC)
- Authors may add a paragraph related to its regulatory part especially clinoril and newly surfaced molecule ipafericept
- Beta-catenin part is very nicely done
- Authors may add APC-beta-catenin signaling cartoon
- Authors may focus on the variety of cancers like CRC, TNBC and ovarian cancer
- Section 5, role of APC and spindle morphogenesis is too lenthy
Round 2
Reviewer 3 Report
Following the revision, it is now a much improved article.
Although it needs little editing before final publication.
Comments:
- Ipafricept is an active molecule in several clinical trials and showing promise in clinical setting with taxane/nab-paclitaxel and gemcitabine in pancreatic cancer and with paclitaxel /carboplatin in ovarian cancer. It is expected from author that he should discuss about this compound more in his article.
- Author may also provide a table related to clinical trial with ipafricept. It will be really helpful for oncologists and also for oncology inclined medical students.
- In line 227, author correct K-RAS to KRAS. Also KRAS is an activating alteration and tP53 is an inactivating alteration.
- Authors also add a small paragraph related to APC-targeted drug. Since APC is a tumor suppressor gene and like other tumor suppressor gene (e.g. TP53 or PTEN), it is not easy to target directly to APC, hence its downstream effector molecule may be a choice for effective treatment approach.
- In line 225 instead of transgenic mice author may re-write to transgenic mouse model...
- In line 223 instead of cancer line, author may write cancer cell line..
- In line 224 instead of wound assay, author may write wound healing assay..
- Part 4 , please make it shorter
- Small typo error are still there (e.g. line 24 missing punctuation
